# Effects of self-administered binaural beats on meditative and introspective states

Micah Amd ⓘ *

Department of Psychology, University of the South Pacific, Suva, Fiji

* micah.amd@proton.me

## Abstract

Emerging evidence suggests select binaural beat (BB) frequencies may enhance regulatory mood states, yet questions remain regarding their efficacy in naturalistic, self-administered settings. This study systematically assessed whether brief (five-minute) self-administered BB exposures modulate regulatory mood states across two studies. In Study 1, 101 participants received 3, 6, 9, or 12 Hz BBs via a 250 Hz carrier. Only theta (6 Hz) BBs significantly enhanced calmness and focus ratings (Hedge's $g$'s > .84; $p$ < .001). Exposure to 9 Hz and 12 Hz BBs significantly enhanced calmness alone ($g$'s > .72; $p$'s < .001). In Study 2, 118 participants underwent 6 Hz BBs, 6 Hz BBs embedded in pink noise, pink noise alone, or silence. Both BB conditions significantly increased calmness and focus ($g$'s > .74; $p$'s < .001). BBs embedded in pink noise were identified to be subjectively preferable relative to isolated BBs, both of which were preferred to pink noise alone or silence ($p$'s < .001). These results provide robust behavioral evidence that brief, self-administered theta BB protocols are effective, accessible, and scalable for enhancing meditative mood states in diverse populations.

## Introduction

Binaural Beat (BB) frequencies describe dissonant 'beat-like' frequencies perceived when two tones of slightly different frequencies are delivered separately (dichotically) to each ear [1]. Although humans can perceive audio frequencies ranging from approximately 20–20,000 Hz, the range for perceived BBs is considerably more constrained [2]. BB detection relies on central auditory processing mechanisms within the brainstem and auditory cortex rather than peripheral cochlear interactions. Central processing of BB frequencies depend on interaural comparison processes constrained by neural conduction delays and synaptic integration times that function to limit temporal resolution capacity [3,4]. These characteristics limit BB perception to carrier frequencies below 1,500 Hz, with BBs optimally perceived between 1–30 Hz [4].

**Data availability statement:** De-identified data, the analysis script and the audio application are available in an online Github repository here (https://github.com/micahamd/BB2-Files).

**Funding:** This study was supported by the University of the South Pacific in the form of an grant awarded to M.A. (AURC 02/2025/1.1.1), and in the form of a research stipend for M.A. The specific roles of this author are articulated in the 'author contributions' section. The funders had no role in study design, data collection and analysis, decision to publish, or preparation of the manuscript.

**Competing interests:** The authors have declared no competing interests exist.

The perception of binaural tones can be considered 'illusory' since the tones presented to each ear do not physically overlap, meaning the experienced dissonance originates from post-auditory processing rather than acoustic interference [5]. Despite this illusory nature, cortically generated BB frequencies have been hypothesized to entrain the brain's oscillatory activity across specific frequency bands, forming the basis of the Brainwave Entrainment Hypothesis, or BEH [6]. The BEH proposes that psychological states associated with particular oscillatory frequencies should become influenced when BBs successfully induce corresponding neural oscillations. On balance, a systematic review by [5] questioned the validity of BEH given inconsistent empirical outcomes, though these were likened to stem from the methodological heterogeneity in EEG measurement approaches used, varying operational definitions of what 'entrainment' entails (auditory steady-state responses versus oscillatory power changes), and inconsistent control procedures across studies.

While the evidence for BEH across all oscillatory bands remains unclear, brainwave entrainment at the 6 Hz (theta) frequency range has been credibly demonstrated by Jirakittayakorn and Wongsawat [6]. Those authors reported notable enhancements in frontal mid-line theta activity within five minutes of exposure to 6 Hz BBs using a 250 Hz carrier tone, with activation intensity peaking within ten to fifteen minutes of exposure and remaining relatively stable for the remainder of the thirty-minute exposure duration. [6] further noted that the increased theta-range activity corresponds with mood states associated with "meditative state(s)," which could be operationally characterized as greater-than-normative levels of calmness and focus. The same 6 Hz BB with 250 Hz carrier setup has also been reported by [7] to reliably modulate event-related potentials (N200, P300), conventionally associated with stimulus discrimination and selective attention, which were behaviorally validated using performance-based measures. On balance, [7] did not report whether actual oscillatory entrainment occurred, nor did they incorporate any measures of participants' subjective experience, making it difficult to determine whether observed effects impacted "meditative" mood states.

A more direct examination was provided by Mendes and colleagues [8], who explored the impact of twenty-minute daily exposures to 6 Hz BBs with a 250 Hz carrier administered over a month across multiple mood states. Those authors reported significant reductions in mood disturbance, which included reduced tension, fatigue, and confusion, alongside increased vigor. These findings provide promising evidence for theta BB exposures facilitating psychological states associated with meditative relaxation and emotional regulation. While Mendes et al. did not report neurological data, they replicated the precise parameter setup shown to entrain frontal midline theta previously [6]. While these studies suggest promise for theta BBs inducing entrainment, it should be noted that [9] and [10] reported no evidence for EEG power enhancement along conventional (theta, alpha, beta, and gamma) frequency bands following BB exposures, contradicting BEH.

The inconsistent outcomes highlight a need for behaviorally validating the practical efficacy of theta BB protocols independently of resolving underlying neurophysiological mechanisms. The present research addresses this gap by focusing on replicable

behavioral outcomes while acknowledging the theoretical uncertainty surrounding underlying neural processes. This work focuses on theta BBs given its increasingly acknowledged role as a functional, and possibly manipulable, bio-marker of mood states associated with cognitive and emotional self-regulation [6,8].

An additional, and understandable, constraint of prior research on theta BBs, at least with respect to confirming brain-wave entrainment and the need to maintain stimulus control, has been the need for participant supervision, given long exposure durations, complex parameter setups and/or the inclusion of psychophysiological measures [5–7,11]. However, for BB entrainment to be "easily accessible and cost-effective," it is paramount to demonstrate whether participants can self-administer established BB parameters effectively. Towards this end, participants in the present study received interactive video tutorials containing detailed instructions on how to self-administer BB auditory stimuli using a custom web application that delivered pure sine waves using the in-built Web Audio API (detailed in Method). The loss of stimulus control that comes with such an approach is offset by a gain in ecological validity, as it evaluates the impact of brief BB sessions in 'real-world' contexts.

The two studies reported here determined whether brief (five minutes) BB sessions could reliably impact mood states relative to control conditions unique to each study. The selection of a five-minute exposure duration, rather than the ten to thirty minutes as reported in earlier studies [6–8,11], was motivated by considerations to minimize potential participant harm while ensuring minimal conditions for entrainment were met. In an unpublished exploratory pilot, several participants reported adverse sensations, including dizziness and head-wringing, following six to eight minutes of continuous BB exposure. Similar negative mood effects were identified across a participant subset by [7], who also identified theta entrainment effects within five minutes of BB exposure. Durations less than five minutes may not reliably produce entrainment [10]. Consequently, a five-minute exposure duration was deemed sufficiently long for influencing theta-states while minimizing potential for task discomfort.

The first study exposed participants to five-minute BB sessions at 3, 6, 9, or 12 Hz, corresponding to conventionally defined delta (3 Hz), theta (6 Hz), and alpha (9, 12 Hz) frequency bands respectively [12]. The comparison of physically equidistant frequencies allowed exploring for any systemic relationships between progressive lowering (or increasing) BB frequencies on relaxation or any other mood states (ramping effects). Additionally, the non-theta BB frequency conditions align with well-established delta and alpha oscillatory activity bands, which have been variably associated with different mood states in prior research [4,5]. Comparing across these conditions would help identify whether (any) significant modulations across mood states were particular to a given frequency.

The second study tested the effectiveness of five-minute 6 Hz BBs against three additional controls: pink noise combined with binaural beats, pink noise alone, and silence. Briefly, 'pink noise' describes a broadband acoustic signal characterized by equal energy per octave, resulting in a power spectral density that decreases at 3 dB per octave. Unlike white noise, which presents all frequencies at equal intensities and can sound harsh, pink noise attenuates lower frequency ranges with interspersed higher frequencies, creating a more natural and less intrusive auditory experience like "the roaring of a waterfall" [11]. The theoretical importance of pink noise and theta BBs derives from [11], who reported that embedding BBs in pink noise inhibits the former's perception while preserving mood-altering effects. This was questioned by [13] however, who suggested pink noise may *amplify* the binaural beat percept and potentially enhance theta entrainment and associated mood states. [13]'s claim is challenged by [5], who reported inconsistent neurological evidence for BB-induced entrainment when binaural beats were embedded in pink noise.

While neurological observations will ultimately confirm or refute the physiological validity of BEH, [6]'s demonstration of neurological entrainment at 6 Hz BBs, coupled with [7] and [8]'s reports of reduced negative moods and enhanced stimulus discrimination abilities respectively using the same parameters, warrants a systematic investigation of pink noise's behavioral potential while maintaining stimulus control. Stimulus control was achieved across the second study by directly comparing 6 Hz BBs presented with or without pink noise backgrounds, relative to groups who received either pink noise exclusively or nothing (silence) under similar conditions.

Alongside mood assessments, participants in the second study completed three open-ended questions assessing their subjectively experienced sensations, level of comfort, and willingness to continue self-administering BB in the future. Qualitative responses were coded as positive, uncertain, or negative by a pair of human raters, providing a comprehensive assessment of the introspective sentiments undermining the different auditory experiences. The second study tested the effectiveness of 6 Hz BBs against three control conditions: pink noise combined with binaural beats, pink noise alone, and silence.

### Research objective

The primary research objective was to determine whether brief, self-administered exposures to theta-frequency (6 Hz) binaural beats reliably enhance core regulatory moods associated with meditative states, such as calmness and focus, in unsupervised online contexts across two studies. The efficacy of theta BBs was compared against 3, 9 and 12 Hz BBs across Study 1. The second study evaluated the effectiveness of 6 Hz BBs presented alone or embedded in pink noise versus pink noise and silence controls, incorporating qualitative participant assessments alongside quantitative change scores. Methodological rigor was maintained via randomization, power calculation, equivalence testing, and manipulation checks to ensure observed effects reflected stimulus-specific mood modulation rather than expectancy or auditory confounds.

## Method

### Participants

**Overview.** All students enrolled in the Psychology program from May 2025 to September 2025 were eligible to participate in the current study. We did not collect specific ethnicity data in compliance with ethical guidelines. No exclusions were made based on demographic, socioeconomic, or other situational factors, as no hypotheses had been specified concerning these variables. All procedures reported in the present study were approved by the University of the South Pacific Ethics Committee on Human Research. All reported participants had provided written consent before commencing their respective tasks. Anonymized identification codes were assigned to individual participants during data collection, meaning they could not be identified afterward. None of the reported studies were pre-registered. Most study participants were incidentally female, reflecting the gender distribution of undergraduate students in the Social Sciences disciplines. Participant ages ranged between 18 and 34 years. All participants had the option of receiving a cash reward (FJ$5.00) following their participation.

**Study 1.** One hundred and eleven undergraduate students enrolled at the University of the South Pacific Social Sciences program volunteered for the first study between June 10th to July 31st, 2025. Ten participants were excluded from the data due to missing responses, leaving one hundred and one participants comprising fifty-five females (24.1 ± 4.9 years) and forty-six males (25.9 ± 5.1 years) that had been randomly assigned to one of four binaural beat frequency conditions: 3 Hz ($N$ = 25; 10 females), 6 Hz ($N$ = 24; 16 females), 9 Hz ($N$ = 27; 10 females), and 12 Hz ($N$ = 25; 19 females). Prospective power analyses on GPower [15] had identified the minimum sample size necessary for identifying a moderate effect (Cohen's $f$ = .25) with 99% power and a 5% error rate for a mixed model comprising a 5-level within-subjects factor (mood-states) and a 4-level between-subjects factor (frequency) was $N$ = 60, keeping the default non-sphericity (epsilon = 1) and auto-correlation ($\rho$ = .5) corrections. Given uncertainty about the latter, additional power analyses were conducted assuming minimum ($\rho$ = .1) and maximum (.9) autocorrelation, which recommended sample sizes of 104 and 16 respectively (Supplementary S1 File in S1 Fig). While our final sample ($N$ = 101) was marginally below the most conservative power estimate ($N$ = 104 for $\rho$ = .1), it substantially exceeded the default power requirement ($N$ = 60 for $\rho$ = .5) and was deemed adequate given the observed effect sizes and the balanced distribution across conditions. The researcher was blind to condition allocation until data analysis.

**Study 2.** One hundred and thirty undergraduate students enrolled at the University of the South Pacific Social Sciences program volunteered for the second study between July 18th and August 5th 2025, using identical recruitment and screening procedures. Sample size was consistent with Study 1's mixed-model power requirements. Twelve participants were excluded for missing responses, yielding a final sample of seventy-two females (27.1±6.1 years), forty-three males (22±2.7 years), and three participants who identified as neither male nor female (22±2.7 years). Participants were randomly assigned to four conditions: BB only (N=34); BB+Pink Noise (N=34); Pink Noise only (N=28); and Silence (N=22). All tasks were completed within twenty minutes. All participants received a cash payment of five Fijian dollars upon completion.

## Materials

**Auditory stimuli.** Two pure sine wave tones were generated using the Web Audio API (*AudioContext.createOscillator*) available across all modern web browsers. The 250 Hz carrier frequency was delivered to the left stereo channel, with group-specific frequencies (247, 244, 241, or 239 Hz) delivered to the right channel to generate 3, 6, 9, or 12 Hz binaural beats, respectively. All auditory stimuli were delivered via headphones worn by the participant, who was responsible for volume adjustment following calibrated instructions. Pink noise was generated using real-time digital filtering approach within the same Web Audio API framework that generated the sine tones. Pink noise samples (uniformly distributed random values between −1 and +1) were processed through a cascade of six first-order infinite impulse response (IIR) filters with predetermined coefficients. This implementation achieved the defining characteristic of pink noise: outputting equal energy per octave with power spectral density decreasing at approximately 3 dB per octave (corresponding to a 1/f frequency response), where higher frequencies are progressively attenuated relative to lower frequencies [11].

**Mood-state surveys.** Participants across both studies completed five 9-point visual analog scales (VAS) that assessed regulatory and hedonic mood states pre- and post-exposure using an interactive online tutorial (Fig 1). These included measures of *Happiness*: "How HAPPY or UNHAPPY do you feel RIGHT NOW?" (1=Very Unhappy, 9=Very Happy); *Calmness*: "How CALM or STRESSED do you feel RIGHT NOW?" (1=Very Calm, 9=Very Stressed); *Peacefulness*: "How AGITATED or PEACEFUL do you feel RIGHT NOW?" (1=Very Agitated, 9=Very Peaceful); *Focus*: "How FOCUSED or DISTRACTED do you feel RIGHT NOW?" (1=Very Distracted, 9=Very Focused); and *Contentment*: "How CONTENT or FRUSTRATED do you feel RIGHT NOW?" (1=Very Frustrated, 9=Very Content). Responses to *Happiness* and *Calmness* scales were reverse scored during analysis.

**Introspective surveys.** Participants across the second study viewed three open-ended questions near the end of the task to catalog their experiences of the task. The three questions were categorized under *Subjective Experience* ("Walk me through your experience from the moment the sound began to when it ended. What specific feelings, sensations, or mental images came to mind?"); *Comfort Level* ("Reflecting on the entire process, how would you describe your overall impression? Were there any aspects that made you feel particularly comfortable or uncomfortable?"); and *Willingness to Repeat* ("Considering the effects you experienced, could you see yourself using an activity like this in your daily life? Please elaborate."). Responses to these questions were coded by two independent human raters, blind to study hypotheses and condition assignments, using a trichotomous sentiment scale (+1=positive, 0=neutral/uncertain, −1=negative). Discrepancies were resolved through structured discussion to reach consensus.

**Study-specific materials.** Participants across both studies underwent a five-minute auditory session sandwiched by mood-state VAS. Participants across the second study additionally completed the three open-ended questions near the end of participation. All survey stimuli were delivered on an interactive video tutorial built using HTML and JavaScript that concurrently guided participants and presented survey items at different phrases. All auditory stimuli were delivered on a self-contained HTML application. All study materials, including the open-source audio application and experimenter transcript, are available in a Github repository (https://github.com/micahamd/BB2-Files).

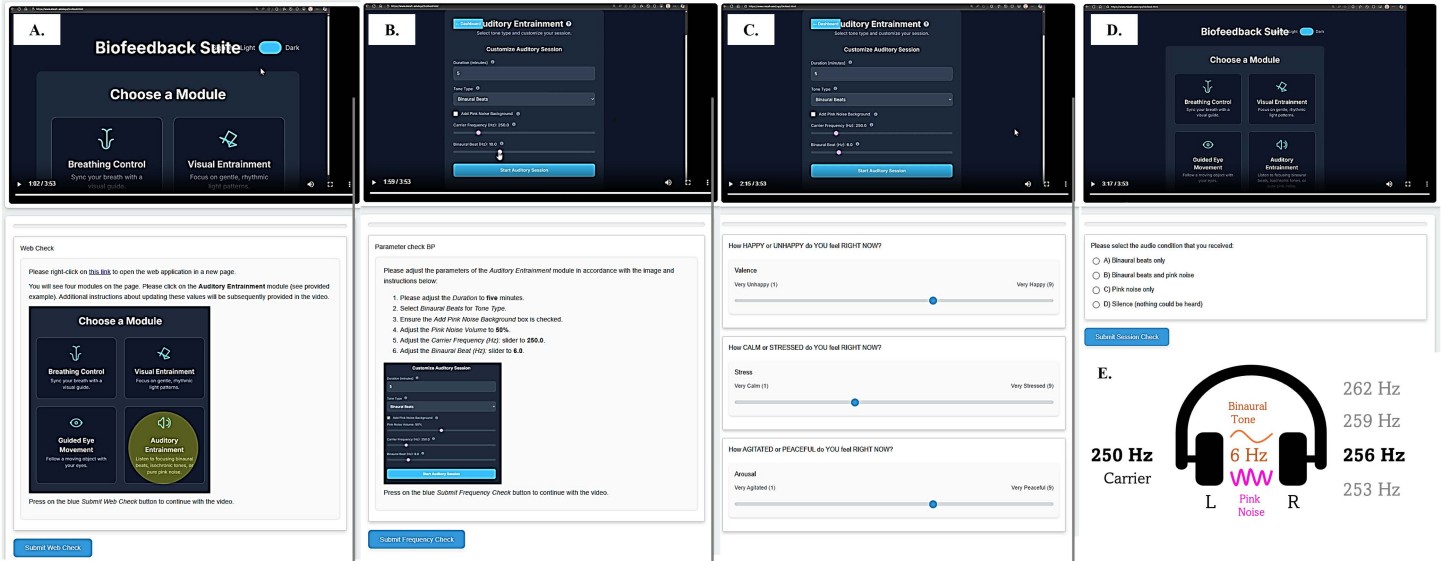

**Fig 1. Schema of task sequence.** (A) All participants viewed an interactive tutorial with instructions to setup the auditory application. (B) Setup instructions were varied based on participant's group affiliation. (C) The auditory phase commenced after participants completed the first mood-state VAS. (D) Participants completed a second VAS and surveys after the audio phase, then asked to re-confirm their group allocation near task's end. (E) All participants received a 250 Hz carrier to the left ear, and a conditional (group-specific) tone to the right ear, the difference corresponding to the target binaural beat frequency. In Study 1, participants were assigned to 3, 6 (theta), 9, or 12 Hz binaural beat (BB) exposure conditions. In Study 2, participants were assigned to theta BB, BB + Pink Noise (PN), PN alone, or silence conditions.

## Procedure

**Study 1.** Participants received a link to the interactive tutorial explaining the upcoming task. Following randomized assignment by the tutorial program to one of four frequency conditions (3, 6, 9 or 12 Hz BBs - see Fig 1), participants completed the first mood VAS, then proceeded to the application hosting the stimulus delivery application. Audio calibration occurred through a standardized protocol: participants pressed a 'Test Audio' button on the interface, which presented 250 Hz tones on both channels for five seconds as participants were instructed to adjust the volume to a comfortable level. This allowed participants to self-modulate loudness to ensure individual comfort. After the audio test, both channels went silent, and participants viewed a message stating the main listening task would commence. Participants received clear instructions to discontinue if experiencing discomfort, though no interventions were required. Post-exposure, participants completed a second mood VAS immediately to capture acute effects before decay. The common task sequence is illustrated in Fig 1. The transcript utilized during the interactive tutorial and survey delivery is provided in the Supplementary S2 and S3 Files.

**Study 2.** Participants received a link to an interactive tutorial, which randomly assigned them to one of four auditory conditions (BB, BB + PN, PN, or Silence). Participants completed mood-associated VAS, then initiated the auditory stimulation after self-modulating their volumes to a pair of 250 Hz tones. Participants allocated to the silence condition also put on the headphones and modulated their volume but commenced the task then with the audio application volume set to 0%. Completion of the five minutes was followed by a second VAS, then the three open-ended questions. Responses to the three questions were typed directly into the text box on the application screen without character or time limits. After responding, participants pressed a 'Submit Results' button, signaling completion.

**Manipulation checks.** Three manipulation checks were implemented to verify participants received the intended auditory stimulation. First, the web application's automatic logging system recorded exact frequency parameters and

exposure duration for each participant, enabling retrospective verification. Second, participants' ability to correctly identify their assigned condition during post-session confirmation (>99% accuracy) suggests successful comprehension. Finally, the application included a time-tracking feature so that any mood-ratings provided in under six minutes would be tagged for disqualification (the minimum required for five-minute stimulus exposure plus completion of both mood assessments). This multi-layered verification system ensured participants received their assigned stimuli for the full duration, though it cannot confirm the subjective perception of binaural beats, which varies across individuals [24].

## Results

### Data preparation

Normalized change scores (Post – Pre/Post + Pre) were calculated for each mood state across each frequency condition, scaling responses to a standardized (−1 to +1) range to mitigate within-subject and inter-item variance [16,17]. Change score distributions for the two studies are illustrated in Fig 2. Descriptive summaries (means and standard deviations) for each mood and frequency combination for the two studies are provided across Tables 1 and 2 respectively. For assumption tests, the full datasets for both studies were fit to linear mixed-effects model objects using the 'lmer' function from the *lme4* R package [18], with participant ID entered as a random factor and Frequency and Mood entered as fixed factors. Shapiro-Wilk tests on model residuals indicated normality was violated for both datasets ($p$'s < .01). Inspection of residual histograms and Q-Q plots indicated influential outliers, which were formally identified as any value exceeding a Cook's distance threshold of $4/n$, with $n$ representing the total observations in the dataset (S4). This commonly used threshold [19] represents a conservative criterion for detecting highly influential observations that may distort parameter estimates. To ensure transparency and robust inference, a second series of datasets were created with outliers removed. We conducted parallel analyses on the full and outlier-removed datasets, consistent with best practices for addressing influential observations [20]. Because Levene's test indicated the full and outlier-removed datasets to be heteroscedastic ($p$'s < .03), all datasets were subjected to an aligned rank transformation using the *ARTools* R package [21,22]. Ranking and aligning data before ANOVA allowed assessing for interaction and main effects, unlike standard non-parametric approaches (e.g., Kruskal-Wallis, Welch's F), while remaining robust to assumption violations like the latter [21]. Significant interactions or main effects were followed by post-hoc contrasts false-discovery rate corrected for multiple comparisons.

### Study 1: Analysis of mood change scores

A Type 3 ART-ANOVA on the full dataset identified a significant main effect for Mood only, $F(4, 388) = 5.508$, $p = 0.022$, Cohen's $f = .17$, with no statistical evidence for a Frequency main effect ($p = .091$), or a two-way interaction ($p = .098$). When the ANOVA was repeated on the outlier-removed data, we observed a significant two-way interaction between Frequency and Mood, $F(12, 362) = 2.521$, $p = 0.003$, $f = .29$, and a significant main effect for Mood, $F(4, 362) = 4.874$, $p = 0.001$, $f = 0.23$. Post-hoc Tukey's HSD tests across the full dataset identified the 6 Hz condition produced significantly larger change scores than the 3 Hz condition overall, HSD [95% CI] = 0.079 [0.011, 0.147], $p = .015$. Across moods, happiness showed significantly less variability than all other mood states (all $p$'s < .007). Probing the interaction confirmed participants in the 6 Hz condition became significantly more focused and calmer relative to participants in the 3 Hz condition (all $p$'s < .002). Post-hoc tests, fdr-corrected for multiple comparisons, across the outlier-removed dataset produced a similar pattern of outcomes. All significant post-hoc tests are detailed in the Supplementary S5 Table. These results demonstrate that the 6 Hz frequency condition produced more substantial mood improvements than the 3 Hz condition, particularly for focus and calmness, with happiness remaining relatively unaffected across conditions. The significant interaction identified in the outlier-removed dataset indicated frequency effects on mood change specifically depending on the specific emotion being measured.

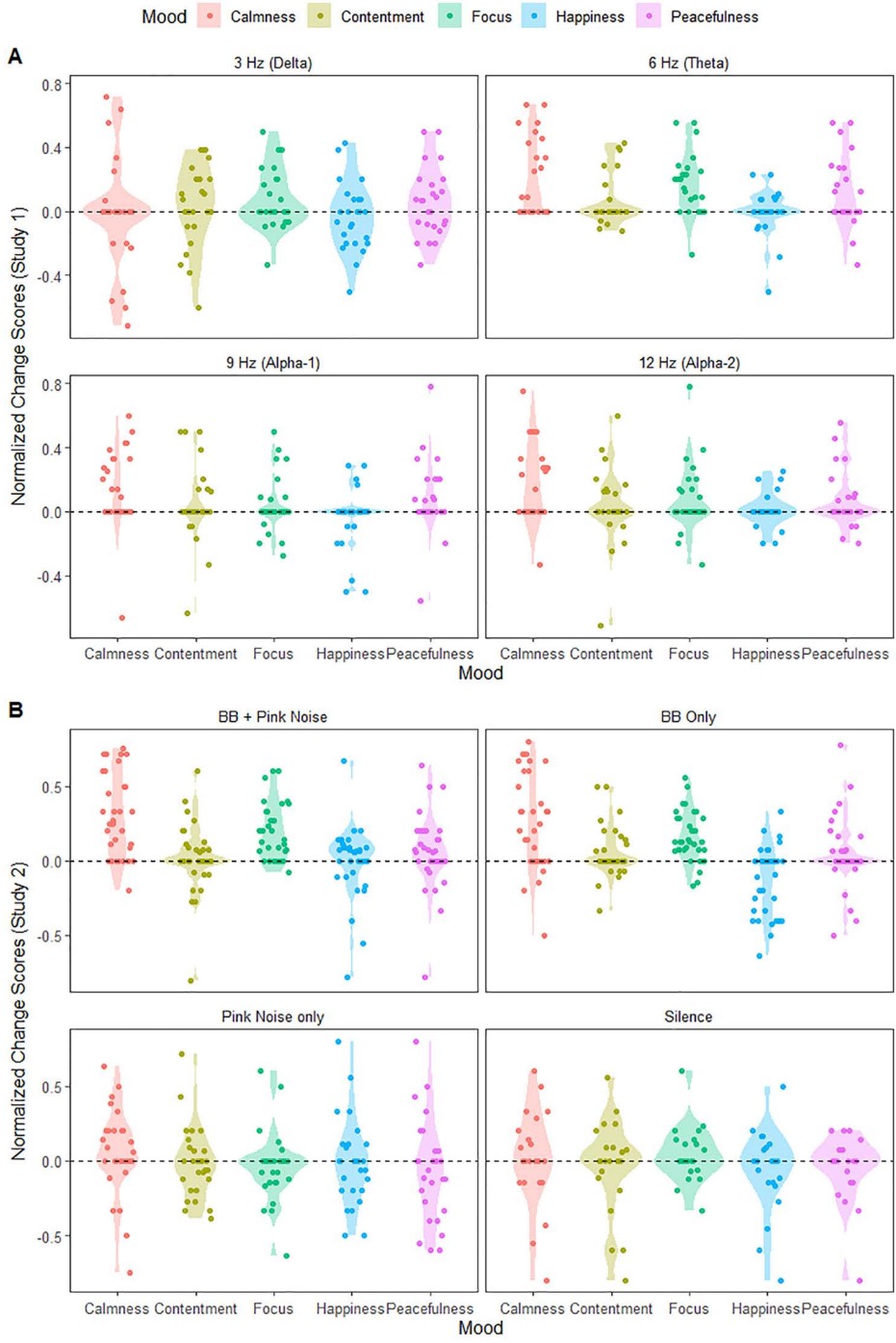

**Fig 2. Violin-plot summaries of mood changes.** Participants change scores (points) across the five moods (x-axis) following different auditory stimulation conditions (facets) across studies 1 (A) and 2 (B). Points were horizontally jittered (width = .1) for legibility.

**Table 1. Change score summaries and effect size estimates of moods across BB groups.**

| Dataset | BB Group | Mood State | Mean | SD | ROPE | Hedge's g |
|---|---|---|---|---|---|---|
| Full | 3 Hz (Delta) | Calmness | −0.02 | 0.35 | ACCEPTED | −0.05 |
| Full | 3 Hz (Delta) | Contentment | 0.04 | 0.25 | ACCEPTED | 0.15 |
| Full | 3 Hz (Delta) | Focus | 0.07 | 0.19 | ACCEPTED | 0.39 |
| Full | 3 Hz (Delta) | Happiness | −0.04 | 0.21 | ACCEPTED | −0.17 |
| Full | 3 Hz (Delta) | Peacefulness | 0.05 | 0.21 | ACCEPTED | 0.25 |
| **Full** | **6 Hz (Theta)** | **Calmness** | **0.22** | **0.25** | **REJECTED** | **0.85** |
| Full | 6 Hz (Theta) | Contentment | 0.07 | 0.16 | ACCEPTED | 0.41 |
| **Full** | **6 Hz (Theta)** | **Focus** | **0.17** | **0.19** | **REJECTED** | **0.85** |
| Full | 6 Hz (Theta) | Happiness | 0.00 | 0.15 | ACCEPTED | −0.03 |
| Full | 6 Hz (Theta) | Peacefulness | 0.12 | 0.23 | UNDECIDED | 0.51 |
| Full | 9 Hz (Alpha-1) | Calmness | 0.14 | 0.25 | UNDECIDED | 0.55 |
| Full | 9 Hz (Alpha-1) | Contentment | 0.04 | 0.24 | ACCEPTED | 0.18 |
| Full | 9 Hz (Alpha-1) | Focus | 0.04 | 0.18 | ACCEPTED | 0.22 |
| Full | 9 Hz (Alpha-1) | Happiness | −0.04 | 0.19 | ACCEPTED | −0.20 |
| Full | 9 Hz (Alpha-1) | Peacefulness | 0.08 | 0.23 | UNDECIDED | 0.35 |
| **Full** | **12 Hz (Alpha-2)** | **Calmness** | **0.18** | **0.25** | **REJECTED** | **0.72** |
| Full | 12 Hz (Alpha-2) | Contentment | 0.03 | 0.24 | ACCEPTED | 0.14 |
| Full | 12 Hz (Alpha-2) | Focus | 0.08 | 0.21 | UNDECIDED | 0.36 |
| Full | 12 Hz (Alpha-2) | Happiness | 0.01 | 0.11 | ACCEPTED | 0.10 |
| Full | 12 Hz (Alpha-2) | Peacefulness | 0.07 | 0.18 | ACCEPTED | 0.37 |
| No-Out | 3 Hz (Delta) | Calmness | 0.00 | 0.14 | ACCEPTED | 0.01 |
| No-Out | 3 Hz (Delta) | Contentment | 0.09 | 0.20 | UNDECIDED | 0.41 |
| No-Out | 3 Hz (Delta) | Focus | 0.06 | 0.17 | ACCEPTED | 0.32 |
| No-Out | 3 Hz (Delta) | Happiness | −0.06 | 0.15 | ACCEPTED | −0.36 |
| No-Out | 3 Hz (Delta) | Peacefulness | 0.02 | 0.17 | ACCEPTED | 0.08 |
| **No-Out** | **6 Hz (Theta)** | **Calmness** | **0.18** | **0.21** | **REJECTED** | **0.80** |
| No-Out | 6 Hz (Theta) | Contentment | 0.07 | 0.16 | ACCEPTED | 0.41 |
| **No-Out** | **6 Hz (Theta)** | **Focus** | **0.19** | **0.17** | **REJECTED** | **1.05** |
| No-Out | 6 Hz (Theta) | Happiness | 0.02 | 0.11 | ACCEPTED | 0.16 |
| No-Out | 6 Hz (Theta) | Peacefulness | 0.10 | 0.17 | UNDECIDED | 0.56 |
| **No-Out** | **9 Hz (Alpha-1)** | **Calmness** | **0.15** | **0.18** | **REJECTED** | **0.85** |
| No-Out | 9 Hz (Alpha-1) | Contentment | 0.01 | 0.13 | ACCEPTED | 0.10 |
| No-Out | 9 Hz (Alpha-1) | Focus | 0.02 | 0.16 | ACCEPTED | 0.15 |
| No-Out | 9 Hz (Alpha-1) | Happiness | 0.00 | 0.15 | ACCEPTED | −0.02 |
| No-Out | 9 Hz (Alpha-1) | Peacefulness | 0.08 | 0.14 | ACCEPTED | 0.56 |
| **No-Out** | **12 Hz (Alpha-2)** | **Calmness** | **0.18** | **0.20** | **REJECTED** | **0.88** |
| No-Out | 12 Hz (Alpha-2) | Contentment | 0.04 | 0.15 | ACCEPTED | 0.28 |
| No-Out | 12 Hz (Alpha-2) | Focus | 0.05 | 0.16 | ACCEPTED | 0.31 |
| No-Out | 12 Hz (Alpha-2) | Happiness | 0.01 | 0.11 | ACCEPTED | 0.10 |
| No-Out | 12 Hz (Alpha-2) | Peacefulness | 0.05 | 0.15 | ACCEPTED | 0.30 |

*Note.* Effects outside the data-derived ROPE indicate statistically significant effects for which the hypothesis of practical equivalence was rejected. Contrasts were run across the full and outlier-removed (No-Out') datasets, respectively. Additional details (e.g., effect size 95% CIs, equivalence, and one-sample *t*-test outcomes) are available in the Supplementary S7 Table in S7 File.

**Table 2. Change score summaries and effect size estimates of moods across audio conditions.**

| Data | Audio Condition | Mood State | Mean | SD | ROPE | Hedge's g |
|------|-----------------|------------|------|------|------|-----------|
| **Full** | **BB+Pink Noise** | **Calmness** | **0.29** | **0.28** | **REJECTED** | **1.01** |
| Full | BB+Pink Noise | Contentment | 0.02 | 0.23 | ACCEPTED | 0.08 |
| **Full** | **BB+Pink Noise** | **Focus** | **0.18** | **0.19** | **REJECTED** | **0.96** |
| Full | BB+Pink Noise | Happiness | 0.00 | 0.24 | ACCEPTED | 0.00 |
| Full | BB+Pink Noise | Peacefulness | 0.06 | 0.25 | ACCEPTED | 0.23 |
| **Full** | **BB Only** | **Calmness** | **0.24** | **0.31** | **REJECTED** | **0.74** |
| Full | BB Only | Contentment | 0.05 | 0.17 | ACCEPTED | 0.31 |
| **Full** | **BB Only** | **Focus** | **0.16** | **0.17** | **REJECTED** | **0.91** |
| Full | BB Only | Happiness | −0.13 | 0.23 | UNDECIDED | −0.53 |
| Full | BB Only | Peacefulness | 0.04 | 0.24 | ACCEPTED | 0.18 |
| Full | Pink Noise only | Calmness | 0.06 | 0.29 | ACCEPTED | 0.19 |
| Full | Pink Noise only | Contentment | −0.02 | 0.24 | ACCEPTED | −0.06 |
| Full | Pink Noise only | Focus | −0.04 | 0.23 | ACCEPTED | −0.15 |
| Full | Pink Noise only | Happiness | −0.01 | 0.29 | ACCEPTED | −0.03 |
| Full | Pink Noise only | Peacefulness | −0.07 | 0.34 | UNDECIDED | −0.19 |
| Full | Silence | Calmness | 0.01 | 0.32 | ACCEPTED | 0.03 |
| Full | Silence | Contentment | −0.04 | 0.32 | ACCEPTED | −0.11 |
| Full | Silence | Focus | 0.04 | 0.19 | ACCEPTED | 0.21 |
| Full | Silence | Happiness | −0.07 | 0.28 | UNDECIDED | −0.22 |
| Full | Silence | Peacefulness | −0.05 | 0.22 | ACCEPTED | −0.20 |
| **No-Outliers** | **BB+Pink Noise** | **Calmness** | **0.23** | **0.16** | **REJECTED** | **1.34** |
| No-Outliers | BB+Pink Noise | Contentment | 0.04 | 0.13 | ACCEPTED | 0.28 |
| **No-Outliers** | **BB+Pink Noise** | **Focus** | **0.22** | **0.14** | **REJECTED** | **1.48** |
| No-Outliers | BB+Pink Noise | Happiness | 0.04 | 0.10 | ACCEPTED | 0.38 |
| No-Outliers | BB+Pink Noise | Peacefulness | 0.06 | 0.12 | ACCEPTED | 0.47 |
| **No-Outliers** | **BB Only** | **Calmness** | **0.21** | **0.13** | **REJECTED** | **1.51** |
| No-Outliers | BB Only | Contentment | 0.06 | 0.13 | ACCEPTED | 0.44 |
| No-Outliers | BB Only | Focus | 0.14 | 0.14 | UNDECIDED | 1.00 |
| No-Outliers | BB Only | Happiness | −0.10 | 0.15 | UNDECIDED | −0.64 |
| No-Outliers | BB Only | Peacefulness | 0.06 | 0.11 | ACCEPTED | 0.51 |
| No-Outliers | Pink Noise only | Calmness | 0.09 | 0.17 | UNDECIDED | 0.51 |
| No-Outliers | Pink Noise only | Contentment | −0.01 | 0.14 | ACCEPTED | −0.08 |
| No-Outliers | Pink Noise only | Focus | −0.01 | 0.18 | ACCEPTED | −0.05 |
| No-Outliers | Pink Noise only | Happiness | −0.01 | 0.14 | ACCEPTED | −0.04 |
| No-Outliers | Pink Noise only | Peacefulness | −0.07 | 0.10 | ACCEPTED | −0.68 |
| No-Outliers | Silence | Calmness | 0.02 | 0.10 | ACCEPTED | 0.20 |
| No-Outliers | Silence | Contentment | 0.00 | 0.12 | ACCEPTED | 0.00 |
| No-Outliers | Silence | Focus | 0.02 | 0.14 | ACCEPTED | 0.11 |
| No-Outliers | Silence | Happiness | 0.02 | 0.11 | ACCEPTED | 0.16 |
| No-Outliers | Silence | Peacefulness | −0.01 | 0.14 | ACCEPTED | −0.07 |

*Note.* Equivalence bounds from the first Study (±.127) were retained for testing practical equivalence.

## Study 1: Testing for practical equivalence

Two one-sided tests for equivalence (TOSTs) were conducted using the *TOSTER* R package [23] to determine whether the observed variance of change scores were practically different from equivalence bounds conservatively estimated from sample characteristics. A sensitivity power analysis across our smallest sample ($N = 24$) had indicated a moderate-to-large effect of Cohen's $d = .597$ could be detected with 80% power and a 5% error threshold for a one-sided contrast. Assuming this as our smallest effect size of interest (SESOI) and multiplying it with the pooled sample standard deviation of the normalized scores ($SD_{pooled} = .212$) generated equivalence bounds (ROPE) of $\pm .127$ appropriate for change scores. The ROPE indicated by these bounds represent the smallest effect size detectable with adequate power in our smallest condition, ensuring that 'null' findings reflect genuinely negligible rather than simply non-significant effects. This approach provides stronger evidence for true absence of effects than traditional null hypothesis testing by enabling three distinct conclusions: rejecting practical equivalence (meaningful effects), accepting practical equivalence (negligible effects), or undecided cases where confidence intervals overlap equivalence boundaries. This approach is a better reflection of the continuum of effect sizes than the binary outcomes output by traditional significance testing [14]. TOSTs run across individual mood states for each frequency condition across the full dataset and across the outlier-removed data are summarized across Panels A and B of Fig 3.

Each TOST is reported with bias-corrected Hedge's *g* estimates across the full dataset and the outlier removed data. Across the full dataset, equivalence tests confirmed that participants allocated to 6 Hz BBs exhibited significantly increased calmness and focus, with participants allocated to 12 Hz BBs also producing significantly increased calmness without overlapping the ROPE, indicating the hypothesis of practical equivalence could be rejected for the former (all Hedge's $g > .71$, all $p$'s $< .001$ – see Table 1). When outliers were removed, the additional finding of significantly increased calmness for participants exposed to 9 Hz BBs ($g = .85$, $p < .001$) was observed, which had been formerly classified as 'undecided' (in the full dataset) due to overlapping the ROPE despite reaching statistical significance ($g = .55$, $p < .001$). The observed effect sizes for significant theta BB effects that rejected practical equivalence ($0.71 > g > 1.05$) substantially exceeded the smallest effect size of interest defined earlier (.597). The TOST outcomes summarized in Table 1 are described in full in the Supplementary S7 Table in S7 File. Equivalence tests confirmed that 6 Hz and 12 Hz frequencies produced practically meaningful increases in calmness, with 6 Hz additionally increasing focus, at levels that substantially exceed ROPE thresholds for practical significance. Removing outliers revealed that 9 Hz also produced a meaningful calmness effect previously masked by data variability.

No corrections for multiple comparisons were applied to equivalence tests, as TOST procedures test for negligible effects rather than differences, and the requirement for two one-sided tests to reach significance provides inherent Type I error control [14]. Additionally, all effects rejecting practical equivalence (Table 1) exhibited *p*-values well below conventional thresholds (all *p*'s $< .001$), suggesting our conclusions remain robust even under conservative correction procedures. The TOST outcomes confirm exposures to 6 Hz produce non-negligible enhancements in calmness and focus, with exposures to 9 Hz and 12 Hz non-negligibly enhancing calmness alone.

## Study 2: Analysis of mood change scores

Across the full data set, a significant two-way interaction was observed between Tone and Mood, $F(12, 456) = 2.054$, $p = 0.019$, $f = 0.23$, with main effects for Tone, $F(3, 114) = 10.67$, $p = 0.001$, $f = 0.53$, and Mood, $F(4, 456) = 8.85$, $p = 0.001$, $f = 0.28$. Re-running the ANOVA on the outlier removed data produced the same pattern of outcomes: a significant two-way interaction, $F(12, 388) = 3.206$, $p = 0.001$, $f = 0.31$, with main effects for Mood, $F(4, 389) = 9.609$, $p = 0.001$, $f = 0.31$, and Tone, $F(3, 110) = 11.185$, $p = 0.001$, $f = 0.55$. Post-hoc Tukey tests, detailed in S6 Table, identified comparable outcomes across the full and outlier-removed datasets. Among groups, the BB Only and BB + Pink Noise conditions produced significantly different change scores relative to Pink Noise Only and Silence conditions ($p$'s $< .01$). Among moods, reports

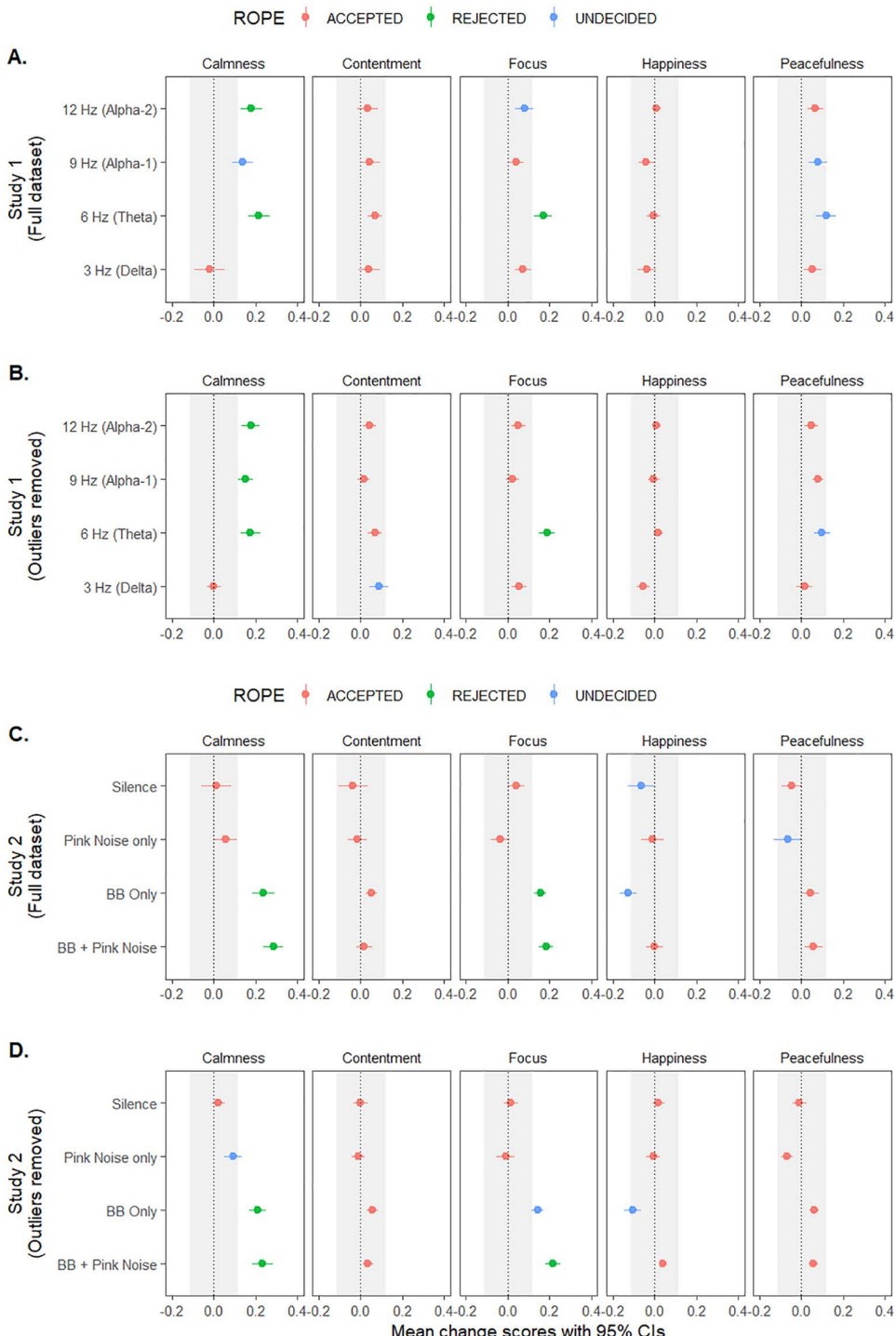

**Fig 3. Mood-change means with 95% confidence intervals (CIs).** Full (A) and outlier-removed (B) data from Study 1, and the full (C) and outlier-removed (D) data from Study 2, with shaded regions illustrating the data-derived equivalence bounds of ±0.127. This comprised the region of practical equivalence, or ROPE. Change scores with CIs outside the ROPE are practically different from zero and non-negligibly significant [(14)]. Exposure to 6 Hz theta BBs, whether presented on their own or with a pink noise background, produced practical and non-negligible increases in self-reported calmness and focus.

for calmness produced significantly different change scores relative to those estimated for contentment, happiness and peacefulness ($p$'s < .01), with focus change scores varying significantly relative to peacefulness scores ($p$ = .043). Probing the significant interactions indicated calmness change scores from the BB + Pink Noise condition was significantly greater than calmness scores from the Silence condition.

These results demonstrate that conditions containing theta BBs, either alone or combined with pink noise, produced significantly greater mood improvements than conditions without binaural beats (pink noise alone or silence). Calmness emerged as the most responsive mood state, showing the largest changes and driving the interaction effect, particularly when BBs were combined with pink noise. The consistency of findings across both full and outlier-removed datasets confirmed the robustness of these effects

### Study 2: Testing for practical equivalence

TOSTs determined whether observed change scores were practically different from the ROPE defined earlier. TOSTs were run across individual mood states for each frequency condition across both datasets (Fig 3C). Change score summaries and bias-corrected effects are provided in Table 2, with additional details (e.g., effect size confidence intervals, equivalence, and one-sample test summaries) available in the Supplementary S7 File. Across both the full ($g$'s > .73, $p$'s < .001) and outlier-removed ($g$'s > .88, $p$'s < .001) datasets, TOSTs confirmed participants allocated to the BB + Pink Noise and BB Only conditions produced significantly greater calmness and focus change scores with variance exceeding the ROPE, indicating the hypothesis of practical equivalence could be rejected (Fig 3D). Interestingly, participants exposed to the BB Only and Silence conditions reported feeling significantly happier across the full and outlier-removed datasets ($g$'s > .5, $p$'s < .004), though these effects overlapped with the ROPE, meaning no decision about their practical equivalence can be made [14]. The Pink Noise Only condition significantly impacted calmness and peacefulness reports ($g$'s > .6, $p$'s < .004), but the hypothesis of practical equivalence could not be rejected. Otherwise, mood-changes for remaining Tone and Mood combinations tested fell within the ROPE.

Equivalence tests confirmed that 6 Hz BBs, whether presented alone or combined with pink noise, produced practically meaningful increases in calmness and focus. While participants also reported increased happiness in the binaural beats only and silence conditions, and increased calmness and peacefulness with pink noise alone, these effects were either too small or too variable to confidently reject practical equivalence. The robust calmness and focus effects associated with 6 Hz BBs replicate the meaningful enhancements in calmness and focus observed across the first study (Fig 3).

### Study 2: Sentiment frequencies

Responses from the three open-ended questions (Q1, Q2, Q3) were coded along a trichotomous (+1/0/-1) sentiment scale (positive/uncertain/negative) by two human raters blind to the study design. For Q1 (SUBJECTIVE_EXPERIENCE), responses were coded as +1 (positive emotional experience), 0 (neutral/uncertain), or −1 (negative emotional experience). For Q2 (COMFORT_LEVEL), responses were coded as +1 (comfortable with the procedure), 0 (neutral/uncertain), or −1 (uncomfortable with the procedure). Q3 (WILLING_TO_REPEAT) responses were coded as +1 (willing to use again), 0 (uncertain about future use), or −1 (unwilling to use again). Raters completed coding independently before comparing results, achieving inter-rater reliability coefficients exceeding. 95 across all three question categories. Discrepancies were resolved through discussion to reach consensus. Spearman correlations across three coded response categories produced positive (.81 > $rho$'s > .46) but non-significant (all $p$'s > .08) coefficients.

Frequency distributions of the trichotomous ratings, depicted in Fig 4, show marked differences in sentiment distributions under different auditory conditions. Chi-square tests explored whether aggregates of positive, uncertain and negative responses were statistically different from chance across each condition. Participants in the BB Only, $\chi^2 (2)$ = 25.9, $p$ < .001, and BB + Pink Noise, $\chi^2 (2)$ = 27.6, $p$ < .001, conditions reported positive experiences significantly more frequently relative to uncertain or negative experiences. Additionally, participants in the Pink Noise Only condition produced the

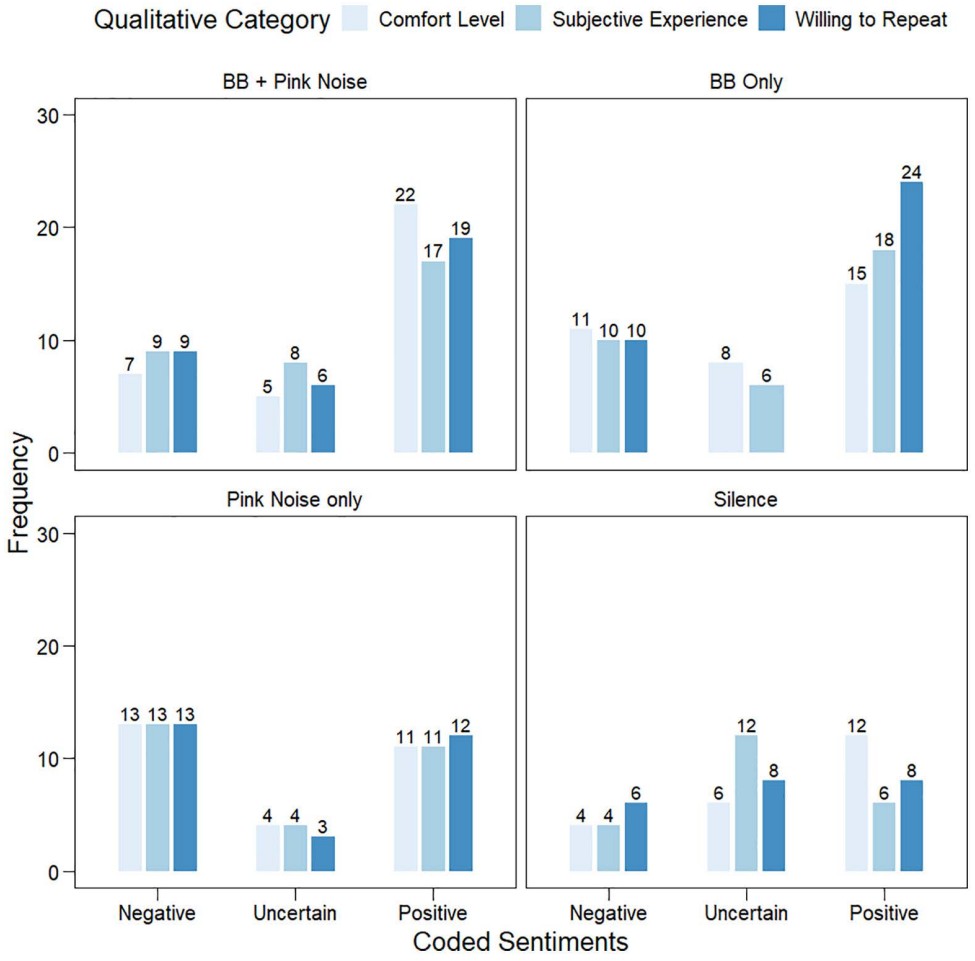

**Fig 4. Frequencies of open-ended sentiments.** Responses describing task comfort, subjective experience, and willingness to repeat were coded as positive, negative, or uncertain (*x*-axis) for the different auditory stimulation conditions in Study 2.

fewest ambiguous responses, $\chi2$ (2) = 15.9, $p < .001$, with coded responses approximately split between negative and positive experiences. Introspective experiences did not significantly vary along positive, uncertain or negative experiences across participants exposed to the Silence condition ($p = .112$). Exploratory correlations between the change scores for the five moods and the coded responses were all positive and mostly (26/28) non-significant. The full correlation matrix is provided in S8.

To directly examine the relationship between quantitative mood changes and qualitative sentiments, five within-subjects ANOVA were run per mood to explore whether coded sentiments covaried with change scores (Fig 5). Significant omnibus effects were found for Calmness, $F$ (2, 351) = 11.93, $p = 0.001$, $\eta^2\rho = 0.064$, for Contentment, $F$ (2, 351) = 22.34, $p = 0.001$, $\eta^2\rho = 0.113$, for Happiness, $F$ (2, 351) = 11.39, $p = 0.001$, $\eta^2\rho = 0.061$, and for Peacefulness, $F$ (2, 351) = 26.11, $p = 0.001$, $\eta^2\rho = 0.13$, but not for Focus, $F$ (2, 351) = 2.03, $p = 0.134$, $\eta^2\rho = 0.011$. Post-hoc tests, detailed in S9, confirmed that negatively coded sentiments corresponded with significantly negative shifts across Calmness, Contentment, Happiness and Peacefulness mood states.

Quantitative analysis of qualitative sentiments indicated significantly higher positive subjective experiences and expressed willingness to repeat the intervention across both BB conditions. Pink noise alone produced polarized

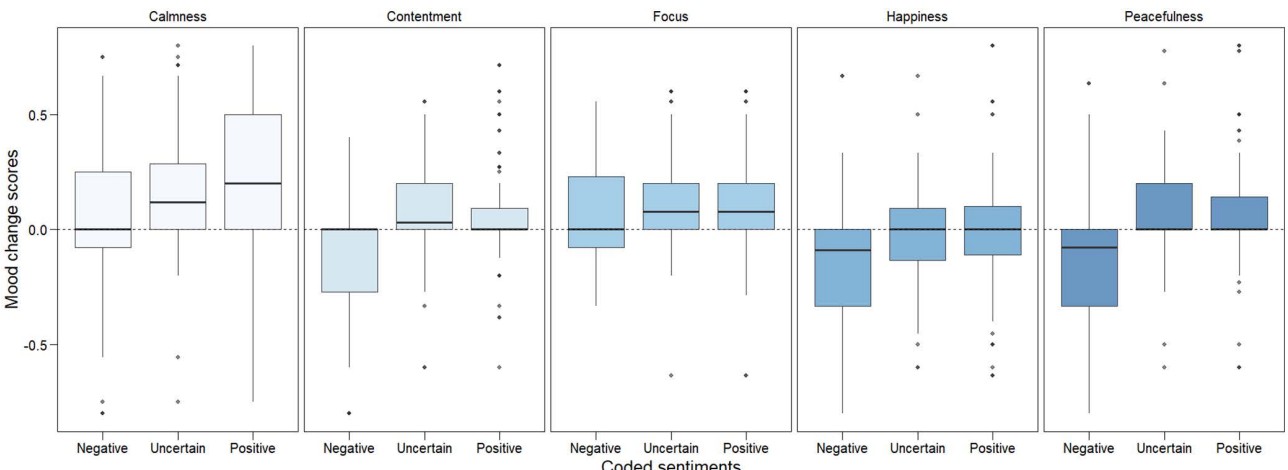

**Fig 5. Change score distributions for sentiments and moods.** Distributions of change scores (y-axis) across coded sentiments (x-axis) for each mood state (panels). Other than Focus (middle panel), negatively coded sentiments significantly co-varied with negative changes in mood states.

sentiments – participants either found it beneficial or aversive, with few in-between responses. The silence condition generated no clear preference pattern, suggesting participants found it neither particularly engaging nor problematic. Subjective sentiments significantly co-varied with most moods other than focus, with exploratory ANOVAs identifying negatively coded introspective statements significantly predicted negative mood shifts.

## Discussion

Across two studies involving over 200 participants, we found compelling evidence that brief, self-administered exposures to theta-frequency (6 Hz) binaural beats can reliably enhance moods associated with meditative states, namely increased calmness, and focus. Study 1 identified that theta (and alpha) entrainment was state-specific, with minimal impact on hedonic indicators such as happiness or contentment. While both theta- and alpha-frequency BBs enhanced calmness, only the former significantly enhanced focus. In Study 2, results practically equivalent to those found following exposures to pink noise or silence confirmed that theta effects were not attributable to generic auditory stimulation, or a lack thereof. However, both BB conditions were comparably effective in inducing calmness and focus, with the inclusion of pink noise rendering the introspective experience more positive. Crucially, most participants across BB conditions indicated willingness to continue BBs on their own, demonstrating the viability and accessibility of the tested protocol for augmenting meditative states.

The relationship between quantitative mood changes and qualitative sentiments revealed a meaningful convergence. Coded sentiments significantly co-varied with measured mood shifts across four of five dimensions (calmness, contentment, happiness, peacefulness), with negative sentiments corresponding to negative mood changes. Focus levels did not co-vary with sentiments ($p = .134$), suggesting participants may not have consciously recognized attentional changes as readily as affective shifts. Across the different auditory conditions, BB presentations produced both significant mood improvements and predominantly positive sentiment frequencies following chi-square analyses ($p < .001$), whereas pink noise alone produced polarized sentiments and mixed mood effects. This multi-method convergence strengthens confidence that theta binaural beats reliably produced positive experiences for most participants, with qualitative data additionally revealing procedural factors (e.g., auditory comfort) and implementation viability (willingness to repeat) not captured by standardized mood measures.

Study 1's unique contribution was comparing physically equidistant frequencies corresponding to established neurophysiological bands (delta, theta, alpha) rather than including a traditional no-stimulation control. Mood-state trends across frequency groups revealed no systematic ramping effects, suggesting BB effects do not follow a linear dose-response pattern across the delta-theta-alpha spectrum. The selective enhancement of regulatory mood states (calmness, focus) rather than hedonic states (happiness, contentment) aligns with established literature linking theta rhythms to frontal executive control networks [6,25]. However, it is important to emphasize that without direct physiological measures (e.g., EEG, fMRI), conclusions regarding neural entrainment remain speculative. The behavioral data reported here cannot inform claims about underlying neural mechanisms, which is a limitation shared by approximately 80% of published BB research that also lack neural measurements [26,27].

Future works could incorporate neuroimaging techniques to better understand the underlying neurophysiological processes during BB exposures [26]. Even at the behavioral level, it could be worth exploring whether other theoretical frameworks may explain observed effects beyond the classic BEH, such as placebo-mediated expectancy effects, attention-arousal interactions, or non-specific relaxation responses to structured auditory stimulation [27]. The frequency-specific pattern observed here (theta > alpha > delta effects) provides some support for entrainment-related mechanisms worth investigating across the proposed studies.

## Limitations

A limitation of the design was a lack of control over potential expectancy effects. It is presently unknown whether any of the participants had prior knowledge about BBs, which could introduce expectancy effects during mood reports [16]. The study was advertised as an investigation of 'auditory stimulation and well-being,' which may have influenced expectancies. While complete blinding is impossible in auditory studies, several design features mitigated systematic bias. First, the frequency-specific pattern of results (theta effects on focus and calmness but not happiness) argues against generalized expectancy effects [28]. If participants simply expected 'auditory stimulation' to improve well-being, we would predict uniform improvements across all measured domains and frequencies rather than the selective theta enhancement of calmness and focus observed. Second, systematic expectancy effects would predict more positive responses in 'active' compared to control conditions, yet 20–46% of participants across all conditions (including binaural beats) reported negative experiences (more on this in a moment). Finally, a lack of robust effects observed for proximal control frequencies (apart from 9 Hz and 12 Hz, which enhanced calmness) suggest stimulus-specific rather than expectancy-driven mechanisms. Future research should implement more sophisticated expectancy controls, including assessment of participants' prior knowledge about binaural beats, explicit expectancy ratings, and potentially active placebo conditions that provide auditory stimulation without hypothesized therapeutic properties [29]. It may also be worthwhile deploying performance-based metrics, such as attentional blink tasks, to functionally validate changes in focus levels [30].

An understandable constraint of prior research on theta BBs, at least with respect to confirming brainwave entrainment and the need to maintain stimulus control, has been the need for participant supervision, given long exposure durations, complex parameter setups and/or the inclusion of psychophysiological measures [6,7,11]. However, for BB entrainment to be "easily accessible and cost-effective" [8], it is important to demonstrate participants can self-administer established BB parameters effectively. Towards this end, participants in the present study received interactive video tutorials with detailed instructions on how to self-administer BBs auditory stimuli using a custom web application that delivered pure sine waves using the in-built Web Audio API (detailed in Method). The loss of stimulus control that comes with such an approach is adequately offset by the ostensible gain in ecological validity by testing the impact of brief BB sessions across naturally occurring contexts.

Another concern worth noting involves the considerable proportion of negative introspective experiences reported across all conditions in Study 2. While several positive responses reflected successfully targeted regulatory states (e.g.,

'Calm and meditation,' 'I feel free' – see S7 Table in S7 File), suggesting mood changes reflected genuine experiential shifts rather than demand characteristics, 20% to 46% of all qualitative responses indicated negative experiences. Inspection of negative sentiments (S7) revealed recurring references to auditory characteristics (e.g., 'The sound was too loud,' 'I felt annoyance at the audio'), highlighting the variability introduced by participant-adjusted volume on outcomes, in spite of the pre-BB audio calibration phase implemented to mitigate this issue. While participant-adjusted volume aimed to enhance ecological validity, it introduced a potential confound in mood measurement that future work should address through standardized protocols or measured covariates. On balance, even silent control conditions elicited negative responses in ~20% of participants, suggesting that individual differences (e.g., in study tolerance) might have contributed to negative experiences over stimulus-specific effects. For instance, individuals with elevated levels of negative affectivity or neuroticism have been reported to evaluate the same experiences negatively relative to participants with lower levels of these traits, even when stimulus conditions are held consistent [31–33]. To address this question, future research should incorporate validated personality assessments (e.g., PANAS – 31) to characterize individual difference moderators and potentially develop personalized approaches to binaural beat administration. Additionally, systematic assessment of auditory processing differences, hearing sensitivity, and prior exposure to similar interventions could inform participant selection and protocol customization.

Finally, the unknown influence of uncontrolled variables (e.g., environmental noise, distractions, task comprehension, headphone quality, volume settings) must be acknowledged as a limitation inherent to any unsupervised protocol. Any self-administered protocol necessarily involves trade-offs between experimental control and ecological validity. Relatedly, the non-supervised protocol did not incorporate screening for the presence of specific hearing deficits, neurological conditions, or medication histories that could affect auditory perception. We acknowledge that these uncontrolled variables (environmental noise, distractions, headphone quality, individual volume preferences, auditory perceptual capacity) introduce measurement error that future protocols should address. Concurrently, several factors support the validity and practical relevance of observed effects.

First, the systematic frequency-specific pattern of results argues that random environmental variability did not produce spurious findings, since statistical 'noise' would be expected to reduce rather than comprise systematic, frequency-specific effects. Second, the web application's built-in calibration procedures and manipulation checks ensured basic protocol compliance. The interactive tutorial included explicit volume calibration procedures using 250 Hz test tones, manipulation checks confirmed appropriate task timing, and any participants completing both mood VAS assessments in under six minutes were automatically disqualified, an event not observed in practice. Third, the ecological validity gained through self-administration is crucial for real-world applications, as future users would similarly self-administer these protocols without external supervision. Nonetheless, future research should systematically examine the boundary conditions of self-administered protocols, including minimal hardware requirements, optimal instructional formats, while ascertaining individual (e.g., personality-related) factors that predict successful self-implementation. Hybrid approaches combining initial supervised training with subsequent self-administration are worth exploring for optimizing intervention protocols.

With respect to user experience, the tutorial and audio application interfaces were designed with accessibility and intuitive interaction as central motivating factors. Clear labels, embedded tooltips, and a mandatory pre-programmed test phase enabled participants to effectively self-calibrate as required. Thorough testing by independent reviewers confirmed the tutorial's effectiveness in effectively conveying task requirements. These design features, coupled with carefully implemented control conditions (e.g., pink noise, silence), strongly suggest that observed outcomes were negligibly affected by procedural variability. Future iterations should empirically evaluate the influence of individual difference variables (e.g., dispositional affect, personality factors) as well as performance-based metrics for mood and attention. Ecological validation through objective real-world outcomes, such as academic performance, will be valuable for determining far-transfer potential beyond self-reports.

## Conclusion

The present work provides a proof-of-concept demonstration that brief, unsupervised 6 Hz BB exposures can enhance theta-associated mood states (calmness and focus) among a geographically dispersed sample of young adults across the South Pacific region (S10). Our findings corroborate the parameters described by [6], which appeared sufficiently robust to induce meaningful change under naturally occurring conditions. Future works are encouraged to examine additional entrainment frequencies and assess behavioral outcomes, leveraging the freely accessible, open-source entrainment web application provided in the online repository for scalable research and deployment.

   (Supplementary files provided after references)

## Highlights

Self-administered exposures to theta (6 Hz) Binaural Beat (BB) frequencies non-negligibly enhanced calmness and focus across two studies.

Mood gains elicited by theta BBs, whether presented in isolation or with a pink noise background, was not replicated across conditions presenting isolated pink noise or silence.

Qualitative assessments identified theta BBs embedded in pink noise to be subjectively preferable relative to isolated BBs, isolated pink noise, or silence.

Most participants expressed willingness to repeat self-administered theta BB protocols, supporting their accessibility and ecological validity for mood enhancement.

## Supporting information

**S1 Fig. Power Analysis Results.** Power analyses for a two-way (4 x 5) ANOVA on GPower with autocorrelation coefficients set to. 1, 5 or 9.
(TIF)

**S2 File. Experimenter transcript.** Prerecorded speech segments presented during different phases of the study.
(DOCX)

**S3 File. Survey Items.** Surveys deployed across studies.
(DOCX)

**S4 Fig. Histograms and QQ Plots.** Histogram and QQ plots of residuals from the linear mixed-effects model for Study 1 (top row) and Study 2 (bottom row). A Cook's distance threshold of 4/n (rightmost column, red intercept) identified significant outliers in the initial model, which were excluded from the outlier-removed datasets.
(TIF)

**S5 Table. Post-hoc test outcomes across Study 1. Significant fdr-corrected Tukey HSD tests on the full and outlier-removed datasets across Study 1.**
(DOCX)

**S6 Table. Post-hoc test outcomes across Study 2.** Significant fdr-corrected Tukey HSD tests on the full and outlier-removed datasets across Study 2.
(DOCX)

**S7 File. Change-scores and TOST outcomes.** Full worksheet in online repository [Worksheet Link].
(DOCX)

**S8 Fig. Correlations across Mood Change Scores and Coded Sentiments.** Correlation matrix across mood-state change scores (peacefulness, calmness, happiness, focus, contentment) and sentiment coded responses to questions about introspective experiences (comfort level, willingness to repeat, subjective experience).
(TIFF)

**S9 Table. Post-hoc tests comparing Mood Change Scores across Coded Sentiments. Post-hoc Tukey tests contrasting distributions of mood change scores across coded sentiment categories.**
(DOCX)

**S10 Fig. Crossbar Summaries of Mood Effects.** Crossbar plots indicating mean changes with 95% confidence intervals (CIs) across moods (y-axis) for the full datasets across Study 1 (row 1) and Study 2 (row 2). Bars with CIs that do not overlap ROPEs (±.127) indicate practically meaningful mood changes (marked).
(TIFF)

## Author contributions

**Conceptualization:** Micah Amd.

**Data curation:** Micah Amd.

**Formal analysis:** Micah Amd.

**Funding acquisition:** Micah Amd.

**Investigation:** Micah Amd.

**Methodology:** Micah Amd.

**Project administration:** Micah Amd.

**Resources:** Micah Amd.

**Software:** Micah Amd.

**Supervision:** Micah Amd.

**Validation:** Micah Amd.

**Visualization:** Micah Amd.

**Writing – original draft:** Micah Amd.

**Writing – review & editing:** Micah Amd.

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
