## [Decision Letter · Decision Letter 0]

29 Nov 2025

Dear Dr. Amd,

Thank you for submitting your manuscript to PLOS ONE. After careful consideration, we feel that it has merit but does not fully meet PLOS ONE’s publication criteria as it currently stands. Therefore, we invite you to submit a revised version of the manuscript that addresses the points raised during the review process.

We look forward to receiving your revised manuscript.

Kind regards,

Francesco Bossi

Academic Editor

PLOS ONE

Journal Requirements:

“The current research was funded by an internal grant (AURC 02/2025/1.1.1) from the USP research office to the author.”

4. Please note that your Data Availability Statement is currently missing the repository name. If your manuscript is accepted for publication, you will be asked to provide these details on a very short timeline. We therefore suggest that you provide this information now, though we will not hold up the peer review process if you are unable.

6.If the reviewer comments include a recommendation to cite specific previously published works, please review and evaluate these publications to determine whether they are relevant and should be cited. There is no requirement to cite these works unless the editor has indicated otherwise.

Additional Editor Comments:

Both Reviewers found it appropriate to ask for minor revisions for the current manuscript. With thorough adjustments and changes, the manuscript can be made available for publication.

Reviewer's Responses to Questions

**Comments to the Author**

1. Is the manuscript technically sound, and do the data support the conclusions?

Reviewer #1: Yes

Reviewer #2: Yes

2. Has the statistical analysis been performed appropriately and rigorously?

Reviewer #1: Yes

Reviewer #2: Yes

3. Have the authors made all data underlying the findings in their manuscript fully available?

Reviewer #1: Yes

Reviewer #2: Yes

4. Is the manuscript presented in an intelligible fashion and written in standard English?

Reviewer #1: Yes

Reviewer #2: Yes

Reviewer #1: The submitted manuscript presents a comprehensive investigation into the behavioral effects of brief, self-administered theta-frequency (6 Hz) binaural beats (BBs) on mood states, specifically calmness and focus, in undergraduate students. The authors report two complementary studies: Study 1 compares frequency-specific effects across delta, theta, and alpha ranges, while Study 2 evaluates the effects of 6 Hz BBs against control conditions (pink noise, silence) and includes qualitative introspective measures. The methodology leverages an online self-administration protocol via a custom Web Audio API application, providing an ecologically valid and scalable approach. Overall, the manuscript is thorough, methodologically rigorous, and addresses a timely question in the field of auditory brainwave modulation. The writing is detailed, and the analyses are statistically sophisticated, incorporating both traditional hypothesis testing and equivalence testing.

In what follows, my comments to the submitted work:

The introduction is dense and highly technical. Simplifying some passages or moving detailed neurophysiological explanations to supplementary material could improve readability.

While the literature review is thorough, the discussion of prior inconsistent findings could be more concise.

Consider briefly justifying why 3 Hz and 12 Hz were included, as readers may question their relevance to delta and alpha ranges.

No screening was reported for hearing deficits, neurological conditions, or use of medications that could affect auditory perception; this should be acknowledged as a limitation.

Participant-adjusted volume introduces variability; discuss potential effects on mood measures.

Clarify rationale for frequency selection and exposure duration in the Methods for ease of understanding.

The Results section is highly technical and dense; providing brief interpretive summaries after each major analysis could improve accessibility.

The connection between quantitative results and qualitative sentiment data could be more explicitly discussed.

It is unclear whether corrections for multiple comparisons were applied across post-hoc and equivalence tests

I would suggest including a summary figure illustrating key frequency/condition effects across moods. Also, consider a concise “take-home message” at the end of each study’s results to highlight the main findings.

Emphasize more strongly that conclusions regarding neural entrainment remain speculative, as no direct physiological measures were collected.

Reviewer #2: Review

The paper investigates the effects of self-administered binaural beats on subjective mood evaluations. The dataset is extensive, and the statistical analyses appear rigorous and appropriate. However, the study relies exclusively on qualitative and self-reported measures, which limits the strength of the conclusions.

Specific comments

1. Abstract – The abstract would benefit from including quantitative outcomes and explicit statistical results to provide readers with a clearer understanding of the magnitude and significance of the findings.

2. Highlights vs. Abstract – There is an apparent inconsistency between the Highlights and the Abstract. The Highlights state: “Mood gains elicited by theta BB stimulation were not replicated across pink noise or silent control conditions.”

However, the Abstract suggests a beneficial role of pink noise: “BBs embedded in pink noise, pink noise alone, and silence. Both BB conditions significantly increased calmness and focus, with a subjective preference for BBs embedded in pink noise reflected by sentiment analysis.”

The authors should clarify this discrepancy.

3. Introduction – The descriptions of Study 1 and Study 2 in the Introduction are unexpectedly detailed and would be more appropriate in the Methods section.

4. Page 1 – The sentence “Study 2: One hundred and thirty undergraduate students … volunteered for the first study” appears to be a mistake. It should refer to the second study.

5. Figure 1 – The text within Figure 1 is too small and difficult to read. Increasing its legibility is recommended.

6. Protocol description – The protocol is described multiple times throughout the manuscript, each time emphasizing different details. A more concise and unified description would improve clarity.

7. Acronym ROPE – ROPE appears several times in figure captions and tables before being defined (on pages 23 and 25). Acronyms should be defined at their first occurrence.

8. Acronym TOST – TOST is defined multiple times. It should be defined once at first use and referenced consistently thereafter.

9. Page 31, line 21 – The phrase “… in practice Third” - A period is missing at the end of the sentence.

**Do you want your identity to be public for this peer review?** For information about this choice, including consent withdrawal, please see our Privacy Policy

Reviewer #1: **Yes:** Gianluca Rho

Reviewer #2: No

---

## [Author Response · Author response to Decision Letter 1]

8 Dec 2025

Response to Reviewers

[Author responses commence with 'Response:...']

Dear Dr. Amd,

Thank you for submitting your manuscript to PLOS ONE. After careful consideration, we feel that it has merit but does not fully meet PLOS ONE’s publication criteria as it currently stands. Therefore, we invite you to submit a revised version of the manuscript that addresses the points raised during the review process. Please submit your revised manuscript by Jan 13 2026 11:59PM. If you will need more time than this to complete your revisions, please reply to this message or contact the journal office at plosone@plos.org. Please include the following items when submitting your revised manuscript:

Response: Attached

Response: Attached

Response: Attached

Response: A Financial Disclosure statement has been included with the updated cover letter.

We look forward to receiving your revised manuscript.

Kind regards,

Francesco Bossi

Academic Editor

PLOS ONE

Journal Requirements:

Response: The correct grant number (AURC 02/2025/1.1.1) has been provided in the ‘Funding Information’ and ‘Financial Disclosure’ sections.

3. Thank you for stating the following financial disclosure: “The current research was funded by an internal grant (AURC 02/2025/1.1.1) from the USP research office to the author.” Please state what role the funders took in the study. If the funders had no role, please state: "The funders had no role in study design, data collection and analysis, decision to publish, or preparation of the manuscript." If this statement is not correct you must amend it as needed. *Please include this amended Role of Funder statement in your cover letter; we will change the online submission form on your behalf.

Response: The following statement has been included in the financial disclosure section: "The funders had no role in study design, data collection and analysis, decision to publish, or preparation of the manuscript."

4. Please note that your Data Availability Statement is currently missing the repository name. If your manuscript is accepted for publication, you will be asked to provide these details on a very short timeline. We therefore suggest that you provide this information now, though we will not hold up the peer review process if you are unable.

Response: The GitHub repository containing all data files (https://github.com/micahamd/BB2-Files) have been included in the Data Availability Statement, and under Study Materials in the revised manuscript.

Response: All data and scripts are available in a GitHub repository (https://github.com/micahamd/BB2-Files)

6.If the reviewer comments include a recommendation to cite specific previously published works, please review and evaluate these publications to determine whether they are relevant and should be cited. There is no requirement to cite these works unless the editor has indicated otherwise.

Response: Understood.

Response: The reference list has been reviewed. No retracted papers are cited.

Additional Editor Comments:

Both Reviewers found it appropriate to ask for minor revisions for the current manuscript. With thorough adjustments and changes, the manuscript can be made available for publication.

Response: We are grateful for the constructive comments and suggestions. All requested edits have been thoroughly administered, which has significantly strengthened the manuscript.

=======

Reviewer's Responses to Questions

Comments to the Author

1. Is the manuscript technically sound, and do the data support the conclusions?

Reviewer #1: Yes

Reviewer #2: Yes

2. Has the statistical analysis been performed appropriately and rigorously?

Reviewer #1: Yes

Reviewer #2: Yes

3. Have the authors made all data underlying the findings in their manuscript fully available?

Reviewer #1: Yes

Reviewer #2: Yes

4. Is the manuscript presented in an intelligible fashion and written in standard English?

Reviewer #1: Yes

Reviewer #2: Yes

5. Review Comments to the Author

Response: All requested changes have been addressed.

Reviewer #1: The submitted manuscript presents a comprehensive investigation into the behavioral effects of brief, self-administered theta-frequency (6 Hz) binaural beats (BBs) on mood states, specifically calmness and focus, in undergraduate students. The authors report two complementary studies: Study 1 compares frequency-specific effects across delta, theta, and alpha ranges, while Study 2 evaluates the effects of 6 Hz BBs against control conditions (pink noise, silence) and includes qualitative introspective measures. The methodology leverages an online self-administration protocol via a custom Web Audio API application, providing an ecologically valid and scalable approach. Overall, the manuscript is thorough, methodologically rigorous, and addresses a timely question in the field of auditory brainwave modulation. The writing is detailed, and the analyses are statistically sophisticated, incorporating both traditional hypothesis testing and equivalence testing.

Response: Thank you for your valuable comments. We hope the revisions address your concerns satisfactorily.

In what follows, my comments to the submitted work:

The introduction is dense and highly technical. Simplifying some passages or moving detailed neurophysiological explanations to supplementary material could improve readability.

Response: The neurophysiological explanations have been reduced by a third in the Introduction on page 4 to avoid unnecessary detail.

1. While the literature review is thorough, the discussion of prior inconsistent findings could be more concise.

Response: Prior literature discussing inconsistent outcomes have been presented more concisely on page 4: "...a systematic review by (5) questioned the validity of BEH given inconsistent empirical outcomes, though these were likened to stem from the methodological heterogeneity in EEG measurement approaches used, varying operational definitions of what ‘entrainment’ entails (auditory steady-state responses versus oscillatory power changes), and inconsistent control procedures across studies."

2. Consider briefly justifying why 3 Hz and 12 Hz were included, as readers may question their relevance to delta and alpha ranges.

Response: Justifications for the frequency conditions (checking for ramping effects, exploring frequency-specific mood effects along conventionally defined bands) have been provided on page 8: "The comparison of physically equidistant frequencies allowed exploring for any systemic relationships between progressive lowering (or increasing) BB frequencies on relaxation or any other mood states (ramping effects). Additionally, the non-theta BB frequency conditions align with well-established delta and alpha oscillatory activity bands, which have been variably associated with different mood states in prior research (4, 5). Comparing across these conditions would help identify whether (any) significant modulations across mood states were particular to a given frequency."

3. No screening was reported for hearing deficits, neurological conditions, or use of medications that could affect auditory perception; this should be acknowledged as a limitation.

Response: The following statement has been included under Limitations section on p. 31: "...the non-supervised protocol did not incorporate screening for the presence of specific hearing deficits, neurological conditions, or medication histories that could affect auditory perception. We acknowledge that these uncontrolled variables (environmental noise, distractions, headphone quality, individual volume preferences, auditory perceptual capacity) introduce measurement error that future protocols should address.”

4. Participant-adjusted volume introduces variability; discuss potential effects on mood measures.

Response: The limitations' section on p. 30 now includes the following: "Inspection of negative sentiments (S6) revealed recurring references to auditory characteristics (e.g., 'The sound was too loud,' 'I felt annoyance at the audio'), highlighting the variability introduced by participant-adjusted volume on outcomes, in spite of the pre-BB audio calibration phase implemented to mitigate this issue. While participant-adjusted volume aimed to enhance ecological validity, it introduced a potential confound in mood measurement that future work should address through standardized protocols or measured covariates...".

5. Clarify rationale for frequency selection and exposure duration in the Methods for ease of understanding.

Response: The rationale for frequency selection is clarified on p. 6: " This work focuses on theta BBs given its increasingly acknowledged role as a functional, and possibly manipulable, bio-marker of mood states associated with cognitive and emotional self-regulation (6, 8)." The rationale for exposure duration is clarified on p. 7: "The selection of a five-minute exposure duration, rather than the ten to thirty minutes as reported in earlier studies (6, 7, 8, 11), was motivated by considerations to minimize potential participant harm while ensuring minimal conditions for entrainment were met…"

6. The Results section is highly technical and dense; providing brief interpretive summaries after each major analysis could improve accessibility.

Response: Brief interpretive summary statements have been included at the end of each Results section on pages 24-28. These include:

- Study 1 ANOVAs on p.24: "These results demonstrate that the 6 Hz frequency condition produced more substantial mood improvements than the 3 Hz condition, particularly for focus and calmness, with happiness remaining relatively unaffected across conditions."

- Study 1 TOSTs on p.26: "Equivalence tests confirmed that 6 Hz and 12 Hz frequencies produced practically meaningful increases in calmness, with 6 Hz additionally increasing focus, at levels that substantially exceed ROPE thresholds for practical significance. Removing outliers revealed that 9 Hz also produced a meaningful calmness effect previously masked by data variability."

- Study 2 ANOVAs on p.27: "These results demonstrate that conditions containing theta BBs, either alone or combined with pink noise, produced significantly greater mood improvements than conditions without binaural beats (pink noise alone or silence). Calmness emerged as the most responsive mood state, showing the largest changes and driving the interaction effect, particularly when BBs were combined with pink noise."

- Study 2 TOSTs on p.28: "Equivalence tests confirmed that 6 Hz BBs, whether presented alone or combined with pink noise, produced practically meaningful increases in calmness and focus. While participants also reported increased happiness in the binaural beats only and silence conditions, and increased calmness and peacefulness with pink noise alone, these effects were either too small or too variable to confidently reject practical equivalence."

- Study 2 Sentiment Analyses on p.

---

## [Decision Letter · Decision Letter 1]

8 Jan 2026

Effects of Self-Administered Binaural Beats on Meditative and Introspective States

PONE-D-25-55053R1

Dear Dr. Amd,

We’re pleased to inform you that your manuscript has been judged scientifically suitable for publication and will be formally accepted for publication once it meets all outstanding technical requirements.

Kind regards,

Francesco Bossi

Academic Editor

PLOS One

Additional Editor Comments (optional):

Reviewers' comments:

Reviewer's Responses to Questions

**Comments to the Author**

Reviewer #1: All comments have been addressed

Reviewer #2: All comments have been addressed

2. Is the manuscript technically sound, and do the data support the conclusions?

Reviewer #1: Yes

Reviewer #2: Yes

3. Has the statistical analysis been performed appropriately and rigorously?

Reviewer #1: Yes

Reviewer #2: Yes

4. Have the authors made all data underlying the findings in their manuscript fully available?

Reviewer #1: Yes

Reviewer #2: Yes

5. Is the manuscript presented in an intelligible fashion and written in standard English?

Reviewer #1: Yes

Reviewer #2: Yes

Reviewer #1: (No Response)

Reviewer #2: All my comments have been carefully addressed by the authors, and the revised version satisfactorily resolves the issues raised in my previous review.

**Do you want your identity to be public for this peer review?** For information about this choice, including consent withdrawal, please see our Privacy Policy

Reviewer #1: **Yes:** Gianluca Rho

Reviewer #2: No

---

## [Editor Report · Acceptance letter]

PONE-D-25-55053R1

PLOS One

Dear Dr. Amd,

I'm pleased to inform you that your manuscript has been deemed suitable for publication in PLOS One. Congratulations! Your manuscript is now being handed over to our production team.

Kind regards,

on behalf of

Dr. Francesco Bossi

Academic Editor

PLOS One